

# The effectiveness of different attentional foci on the acquisition of sport-specific motor skills in healthy adults: a systematic review with network meta-analysis

Emmanuel Favre-Bulle[1,*], Siri Nyfeler[1,*], Chloé Schorderet[1,2], Gaia Risso[1,2], Michela Bassolino[1,2] and Karl Martin Sattelmayer[1]

[1] Institute of Health Sciences, School of Health Sciences, HES-SO Valais-Wallis, Leukerbad & Sion, Switzerland
[2] The Sense Innovation and Research Center, Sion & Lausanne, Switzerland
* These authors contributed equally to this work.

## ABSTRACT

**Background:** The acquisition of motor skills is a key element in many sports. A motor learning principle, which is frequently used to support skill acquisition is the application of different attentional foci. The effectiveness of different attentional foci on performance and the learning of motor skills has been investigated in various sports using randomised controlled trials. The aim of the present study was to investigate the effectiveness of different attentional foci (such as external (EFA) and internal attentional foci (IFA), but also holistic and switching foci) on the performance and learning of a sport-specific motor task in healthy individuals.

**Methods:** This study was a systematic review with network meta-analysis. We followed the Prisma reporting guideline and the Cochrane handbook for systematic reviews. Cinahl, Embase, Medline and Cochrane Central were searched for eligible studies. Network meta-analyses were performed for the post-acquisition, retention and transfer test endpoints.

**Results:** Twelve studies were included in the review. At post-acquisition an EFA was the most effective intervention compared to the control intervention (SMD: 0.9855; 95% CI [0.4–1.57]; $p$: 0.001). At the retention and transfer test endpoints, a holistic focus of attention had the highest effectiveness compared to an IFA (SMD 0.75; 95% CI [−0.1 to 1.6]; $p$: 0.09) and (SMD 1.16; 95% CI [0.47–1.86]; $p$: 0.001).

**Discussion:** For all three endpoints, we analysed a greater effectiveness of an EFA and holistic focus compared to an IFA. Several promising different attentional focus interventions were identified. The largest effects were analysed for a holistic focus. However, only one study used this intervention and therefore there remains uncertainty about the effectiveness. With regard to the inconsistency observed, the analysis at post-acquisition should be interpreted with caution. Modified versions of the EFA were the imagined and the dynamic EFA. Both were only explored in single studies and should therefore be investigated in further follow-up studies that directly compare them.

Corresponding author
Karl Martin Sattelmayer,
martin.sattelmayer@gmail.com

## INTRODUCTION

The acquisition of motor skills is a key element in many sports. Learning has been defined and differentiated from "performance" by *Magill & Anderson (2010)*. They suggest that learning is indicated by a reliable and enduring improvement in skill execution, which is generally the result of prolonged and repeated practice (*Magill & Anderson, 2010*).

Performance on the other hand is defined as "observable behaviour" (*Magill & Anderson, 2010*, p. 257). A variety of tests are available for the assessment of motor skill acquisition (*Wulf, Shea & Lewthwaite, 2010*). Assessment of performance immediately after a period of training is often referred to as post-acquisition testing. The results of these tests may indicate that a particular training intervention has changed the trainee's ability to perform a particular motor skill, but given the short period between acquisition and testing, it is difficult to assess whether motor learning has occurred (*Schmidt & Lee, 2014*). The learning of a motor skill can be assessed with retention and transfer tests.

In retention tests, a retention period is set between the acquisition phase and the actual test. The length of the retention period is defined differently in the literature, but a minimum of 24 h is often used. Transfer tests are used to assess the generalisability of what has been learned. Participants are asked to perform a motor skill that is similar to, but different from, the skill that has been practiced. There can also be a change in conditions (*Schmidt & Lee, 2014*). For example, *Hadler et al. (2014)* used a test position on the right side of a tennis court as a transfer test for children learning a tennis task. During acquisition, the motor skill was instructed from the centre of the court (*Hadler et al., 2014*). An increase in performance on retention or transfer tests is an indication of learning (*Magill & Anderson, 2010*)

Several principles have been suggested, which can be used to increase the effectiveness of motor skills acquisition (*Magill & Anderson, 2010*). One motor learning principle, which is frequently used in practice is the application of different attentional foci. The principle was originally investigated by *Wulf, Höß & Prinz (1998)* and involves comparing an internal focus of attention (IFA) (*i.e.*, instructions are directed towards the learner's own body) with an external focus of attention (EFA) (*i.e.*, attention is directed towards the effects of the learner's actions on the environment).

*Wulf, McNevin & Shea (2001)* proposed the "constrained action hypothesis" which states that an IFA involves conscious control of motor actions and an EFA involves more automatic motor control processes. It has been stated that conscious control of motor activity may limit or interfere with the normal automatic processes that are involved in the learning of motor skills.

The effectiveness of different attentional foci on performance and the learning of motor skills has been investigated in various sports using randomised controlled trials. For example, the randomised controlled trial of *Abedanzadeh, Becker & Mousavi (2022)* investigated the effects of different attentional foci on the performance and learning of a badminton short serve. Instructions in the EFA group where: "focus on the movement of the racket during the serve". Participants in the IFA group received the following instructions: "focus on the movement of your arm during the serve". The results suggested

an advantage of an EFA over an IFA when learning this motor task. Another example of the application of different attentional foci to the acquisition of sport skills is the study by *Land, Frank & Schack (2014)*. The authors investigated if focus of the instructions influenced the acquisition of a golf putting task. The participants were divided into two groups where the instructed attentional focus was either on the movement of the arm swing (IFA) or on the speed of the ball (EFA). Their findings suggested that an EFA is more effective for the acquisition of a golf putting task.

Two systematic reviews have been published on the effectiveness of different attentional focuses on the acquisition of motor skills (*Chua et al., 2021*; *Makaruk, Starzak & Porter, 2020*). The aim of the meta-analysis by *Makaruk et al. (2019)*, was to compare the effectiveness of different instructions (EFA, IFA and a control group (*i.e.*, participants received instructions without an attentional focus such as "jump to the best of your ability" in *Asadi et al. (2019)*) in performing vertical and horizontal jumps at a post-acquisition test. In total 14 studies were included into their analyses. The results showed that the overall jump performance was better with the EFA instructions than with the IFA instructions or in the control group.

The meta-analysis by *Chua et al. (2021)* investigated the effect of an EFA compared to an IFA on motor performance and motor learning. They included 88 studies for motor performance and 52 for motor learning). Their analysis showed an advantage in favour of the EFA. Their multivariable meta-regression showed that an EFA is superior to an IFA and that neither age, health status nor ability level changed the superiority.

However, research over the last two decades in sports has shown that the use of attentional focus strategies is likely to be more complex than just using an EFA or an IFA. More recently, researchers such as *Abedanzadeh, Becker & Mousavi (2022)* have proposed other strategies going beyond the dichotomous use of attentional foci. An example of this is the use of a holistic-attentional focus, which can be described as concentrating on the overall sensations or emotions associated with performing a movement (*Becker, Georges & Aiken, 2019*). The holistic-attentional focus, showed benefits in the performance and learning of an accuracy-based task, compared to an IFA or a control group (*i.e.*, no specific focus cue was given during the instructions) (*Abedanzadeh, Becker & Mousavi, 2022*). Another attentional focus strategy, which have been proposed by *Aiken & Becker (2022)* is a switching focus of attention. For instance, participants in the switching group used an IFA during preparation and an EFA during execution of a motor skill. Their results suggest that the negative impacts of an IFA emerge only during skill execution and not during its preparation. This would suggest that learners can potentially gain advantages from both IFA and EFA as long as these cues are introduced at the appropriate stage of skill performance.

We believe that a systematic review is missing for the following reasons: i) the current level of evidence is based on pairwise comparisons. This means that evidence of effectiveness is only available from head-to-head comparisons (*Jansen & Naci, 2013*), such as EFA *vs* IFA or IFA *vs* control, ii) the comparisons are restricted to the interventions EFA, IFA and control. Primary studies also compare other types of attentional foci, and the

effectiveness of these interventions is not currently analysed, iii) the existing systematic review have investigated either a very specific motor skill (*i.e.*, jumping performance (*Makaruk, Starzak & Porter, 2020*)) or a very broad range of motor skills (*e.g.*, a variety of motor skills not only related to motor skills in sports (*Chua et al., 2021*)). An evaluation of sport-related motor skills is therefore missing, iv) there is no evidence available from network meta-analyses.

A network meta-analysis is a statistical approach that facilitates the simultaneous evaluation of various treatments by merging both direct and indirect data from numerous studies. This method is particularly advantageous when multiple treatments or interventions exist for a specific outcome, as it allows for the determination of each treatment's comparative effectiveness within the network, including those not directly contrasted in comparative studies (*Chaimani et al., 2023*). Direct data refers to evidence retrieved from studies comparing specific interventions. In the context of this review, study A would be a comparison between an EFA and an IFA instruction, while study B would be a comparison between an EFA and a control instruction. This set of studies would provide direct evidence for the comparison of EFA *vs* IFA and EFA *vs* control. Indirect evidence would be available for the comparison IFA *vs* control. There are several advantages associated with network meta-analyses: i) the additional integration of direct and indirect evidence leads to an increase in the precision of effect estimates compared to traditional meta-analyses (*Chaimani et al., 2023*). From our perspective, this enhanced precision is of paramount importance in disciplines such as motor learning, where even minor discrepancies in efficacy can have significant practical implications; ii) integration of multiple comparisons. Unlike the existing pair-wise meta-analyses (*Chua et al., 2021*; *Makaruk, Starzak & Porter, 2020*) a network meta-analysis allows the simultaneous comparison of multiple attentional foci interventions, offering a more comprehensive understanding of their relative effectiveness (*Jansen & Naci, 2013*); iii) with the help of the network estimates it is possible to establish a hierarchy of interventions. This represents a significant advance over pairwise meta-analyses, in that it provides guidance to practitioners on the most effective attentional focus intervention for motor skill acquisition.

Therefore, the aim of the present study was to investigate the effectiveness of different attentional foci (not only considering EFA and IFA) on the performance and learning of a sport-specific motor task in healthy individuals.

## MATERIALS AND METHODS

### Design

This study was a systematic review with network meta-analysis. We followed the guidelines of the Cochrane Handbook for Systematic Reviews of Interventions (*Higgins et al., 2022*) and structured the reporting according to the PRISMA statement for network meta-analyses (*Hutton, Catala-Lopez & Moher, 2016*). A protocol of this review was published in the OSF registries (*Favre-Bulle et al., 2022*).

## Searches

The search was performed on four electronic bibliographic databases; CINAHL Complete, Embase, MEDLINE (OVID) and Cochrane CENTRAL. The search strategy was developed and finalised on MEDLINE and then applied to the other databases mentioned above. Three search concepts were used to construct the search strategy. These comprised the i) population (*i.e.*, people learning or training a sport-specific motor task), ii) interventions (*i.e.*, different attentional foci), iii) outcomes (*i.e.*, outcomes used to assess performance and learning) and a search filter for RCTs (*Higgins et al., 2022*). For each of the search concepts multiple free text search terms were identified and combined with subject headings (*i.e.*, MESH terms). The search strategy for MEDLINE is presented in Appendix A1.

## Condition or domain being studied

Acquisition of a sport-specific motor task.

## Participants

Studies needed to report on adults (age between 18 and 65 years) learning or training a specific sport. There were no restrictions on the type of sport task or the level of experience in the sport. Several exclusion criteria were applied and the following populations were excluded: i) individuals with an injury or chronic illness, ii) individuals undergoing rehabilitation, iii) individuals over the age of 65 and minors.

## Interventions

The sport-specific task had to be trained using at least one specific focus of attention. Multiple attentional foci could be compared within a study. Examples of eligible attentional foci were an internal or external attentional focus. In IFA, attention is directed to the body, whereas in EFA the attentional focus is directed to the movement effect (*Wulf, 2013*). This systematic literature review also included studies investigating other attentional foci such as a mixed attentional focus. A mixed attentional focus may comprise a combination of internal and external focused instructions. For example, an IFA is used in the preparation phase and an EFA is used in the execution phase of a motor skill (*Connor et al., 2019*).

## Comparator

The included studies needed to have a control group. The control group could have a specific focus of attention, but another control intervention or a sham intervention was also possible.

Only studies with a randomized controlled trial design were included. Studies based on a cross-over design were also eligible for inclusion under two conditions: i) there was a random group allocation and ii) participants received only the allocated intervention for the specified cross-over period. For example, studies were excluded, if participants practiced a task with different attentional foci in the same period.

## Main outcomes

The main outcomes of this review were the performance and learning of a specific sport-specific motor skill. Different outcome measures were allowed, as different sports were included. If a study reported multiple outcome measures for a sport-specific task, the review team reached a consensus on which outcome measure to use. The following factors played a role in the decision; i) homogeneity in terms of outcome measures of the included studies and ii) which outcome measure most accurately measured the performance and learning of the trained motor task. In order to prevent a unit-of-analysis error in our meta-analysis, it was determined that a single outcome measure should be included per study. In particular, the objective was to prevent the double counting of studies (*Senn, 2009*), ensuring that each participant's data is represented only once in any given meta-analysis for each outcome assessed. Consequently, the most suitable outcome measure was selected for each study endpoint.

Performance at post-acquisition was the main outcome of this review and was assessed immediately after the intervention (*i.e.*, training period) using post-acquisition tests. Two additional endpoints were analysed as indicators of learning. First, performance was assessed with retention tests. We used a minimal retention period of 24 h between the training period and the assessment of performance. In addition, transfer tests were used to analyse the performance on different tasks or conditions (*Schmidt & Lee, 2014*).

## Data selection

The study selection was carried out in three steps. 1) All identified datasets were searched electronically for duplicates using the application Zotero (*Guimarães et al., 2022*); 2) The titles and abstracts of the remaining datasets were screened independently by two reviewers (SN, EFB). This step was performed using the software RAYYAN (*Ouzzani et al., 2016*). In a final step, the full text articles of the remaining datasets were screened independently by two reviewers (SN, EFB). In case of disagreements, the decisions were discussed jointly and, if necessary, resolved by a third person (KMS or CS).

## Data extraction

Two examiners (SN, EFB) independently extracted relevant data into an electronic data extraction form. In case of missing data, the authors of the primary study were contacted by mail and asked for the data concerned. If data was not available in numeric form we used the application "WebPlotDigitizer" to extract data from figures (*Drevon, Fursa & Malcolm, 2017*).

## Assessment of methodological quality

The PEDro scale was used to evaluate methodological quality of the included studies (*de Morton, 2009*). This process was carried out by two independent reviewers (SN, EFB). Conflicts were resolved by discussion within the review team.

## Strategy for data synthesis

The data synthesis followed the guidelines of the Cochrane Handbook for Systematic Reviews of Interventions (*Higgins et al., 2022*). The primary endpoint for assessing the effectiveness of the comparisons was at the end of the intervention (*i.e.*, performance measured with post-acquisition tests). Secondary analyses were conducted using data from i) the longest available retention test and ii) transfer tests.

## Analysis of the data

The analysis was performed using the statistical software R (*R Core Team, 2023*). A frequentist network meta-analysis was performed using the R package "netmeta" (*Rücker et al., 2023*). The transitivity assumption was examined for all studies included in the network meta-analysis (*Salanti et al., 2014*). The transitivity assumption is essential for network meta-analyses and the incorporation of indirect comparisons. Indirect observations provide observational evidence, but these may also contain biases and confounding. In order for an indirect comparison to be considered valid, it is necessary that the included studies exhibit, on average, comparability in all significant aspects, with the exception of the specific interventions being compared (*Chaimani et al., 2023*). In a more practical sense, studies that provide direct evidence (A against C and B against C) should exhibit a similar distribution of key aspects, including participants' characteristics and methodological aspects. It is only then that the evidence of the indirect comparison (A against B) can be considered valid (*Chaimani et al., 2017*). Within this study, this was checked in two stages: Firstly, we investigated whether important effect modifiers were sufficiently similar between studies. For this we checked whether the following characteristics were similarly distributed: skill level at baseline, sample size and methodological quality measured with the Pedro scale. Secondly, we checked whether all interventions were meaningful to all participants. This means that any participant could theoretically have received any treatment in the network of interventions (*Chaimani et al., 2017*).

Regarding the relatively high degree of heterogeneity of included interventions, a random effects model was used for the network meta-analysis. The network of interventions was visualised using network plots. Within the plot, each intervention is represented as a circle. Circles are connected if there was at least one pairwise comparison. The network was analysed with regard to the presence of open and closed loops. The term "closed loop" is employed to describe comparisons with multiple pathways, including direct and indirect evidence. The term "open loop" is employed to describe comparisons that do not include direct evidence (*Antoniou et al., 2019*). Intervention effects were analysed as standardised mean differences (SMD). Effect sizes were classified as suggested by *Cohen (1992)*. That is, 0.2 was considered a small, 0.5 a moderate and 0.8 a large effect. Effect estimates were presented in forest plots. We set the comparator to the intervention with the lowest estimated effectiveness. Direct and network estimates were compared and tabulated in a netleague table. P-Scores were used to produce a treatment ranking of the included interventions (*Rücker & Schwarzer, 2015*). P-Scores were calculated based on the

observed effect sizes and corresponding standard errors, ranging from 0 to 1 on a continuous scale, with higher values indicating higher rankings. Heterogeneity was explored using Higgins' I2 statistics and classified using the reference values from the Cochrane Handbook for systematic reviews of interventions (*Higgins et al., 2022*). In addition, we explored heterogeneity within designs and inconsistency between designs. Furthermore, inconsistency was visualised using a net heat plot or if not available with net splitting (comparison of direct and indirect effect estimates). A possible publication bias (*i.e.*, the chance that studies with statistically significant findings are more likely to be published) was explored with comparison-adjusted funnel plots (*Salanti et al., 2014*).

## Sensitivity analyses

Heterogeneity was explored with the help of meta-regressions. We used the meta package for these analyses (*Schwarzer, Carpenter & Rücker, 2015*). In a first step we performed pair-wise meta-analyses of the comparisons with the largest heterogeneity. A random-effects model was employed, with the restricted maximum likelihood ratio selected to estimate heterogeneity variance. In the second step, a mixed-effects model was utilised for the meta-regressions. We explored the influence of the variables age of participants, number of trials performed during the acquisition phase and methodological quality (*i.e.*, measured with the number of items on the PEDro scale) on the effect estimates of the meta-analyses. Furthermore, we investigated the impact of removing studies with disparate effect sizes on the heterogeneity in the network meta-analysis.

# RESULTS

## Findings of the search

The search on the four databases resulted in a total of 5,031 identified records. After exclusion of duplicates, 3,952 records remained, which were screened for title and abstract. Of these, 3,872 records did not meet the criteria for further processing. Of the resulting 80 records, the full texts of 79 were identified. Subsequently, the reviewers screened the full texts for the specific inclusion and exclusion criteria, excluding 67 records. Finally, 12 studies were included in the review. The study selection process is shown in Fig. 1.

## Included studies

A total of 566 subjects were included in the systematic review. Badminton (*Abedanzadeh, Becker & Mousavi, 2022*), golf (*Aiken & Becker, 2022*; *Land, Frank & Schack, 2014*; *Munzert, Maurer & Reiser, 2014*; *Poolton et al., 2006*; *Wulf, Lauterbach & Toole, 1999*), basketball (*Al-Abood et al., 2002*; *Benjaminse et al., 2017*), volleyball (*Rostami et al., 2020*), standing long jump (*Yamada, Raisbeck & Porter, 2021*), soccer (*Makaruk et al., 2019*), and dart throwing (*Marchant et al., 2009*) were the sports trained. The attentional focus strategies used in the study interventions were: EFA, IFA, holistic focus and switching focus of attention. In addition, there were variations of an EFA such as a dynamic EFA (*i.e.*, instructions were used to focus the attention on the movement dynamics of a video model) and an imagined EFA (*i.e.*, instructions were focused on an imagined EFA). All interventions and intervention labels are presented in Table 1. A post-acquisition endpoint

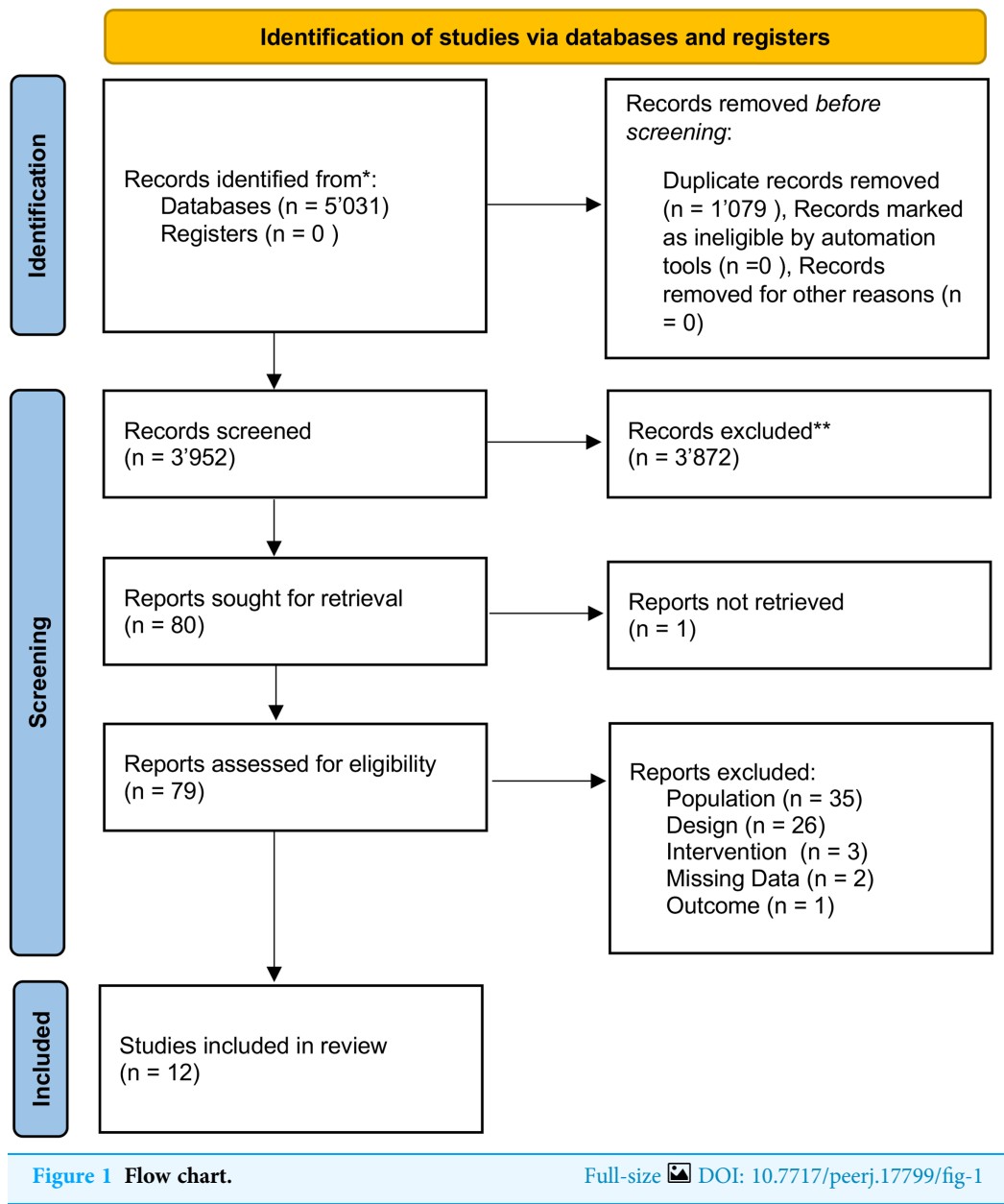

**Figure 1 Flow chart.**

was administered in all experiments. The endpoint retention test was performed in nine experiments. A transfer test was identified in six experiments. All study characteristics are listed in Table 2.

## Methodological quality

For the following PEDro items, all studies included in the syntheses were considered at low risk of bias: random allocation, baseline comparability, outcomes of >85% of subjects, intention-to-treat analysis, between group comparison. Apart from *Rostami et al. (2020)*, all studies received a high risk of bias for concealed allocation and blinding of therapists. In addition, all studies had a high risk of bias for the criterion subject blinding. The methodological quality is shown by means of a percentage diagram in Fig. 2.

**Table 1 Interventions and labels.**

| Intervention | Label | Description of intervention |
|---|---|---|
| Control | Control | A control instruction without a specific focus during the instructions |
| Dynamic external focus of attention | Dynamic EFA | The instructions are focussed on the movement pattern of a model |
| External focus of attention | EFA | The attention is directed towards the effects of the learner's actions on the environment |
| Holistic focus of attention | Holistic focus | The attention is focussed on the overall sensations and emotions associated with the movement |
| Internal focus of attention | IFA | The instructions are directed towards the learner's own body |
| Imagined external focus of attention | Imagined EFA | As for IFA with the exception of an imagined object or effect |
| Switching focus of attention | Switching focus | The focus of attention changes during the instructions (*e.g.*, internally during preparation and externally during execution) |

## Analysis of effectiveness

To test for transitivity, we checked whether potential effect modifiers (*i.e.*, sample size, skill level and study quality) were similarly distributed across the comparisons. For all comparisons the sample size was <100 participants (mean sample size was 52 participants), the smallest average sample size was 45 (EFA *vs* holistic, IFA *vs* holistic) and the largest average sample size was 79 (EFA *vs* switching focus, IFA *vs* switching focus). The majority of participants were novices in all comparisons. For study quality, the average number of items with problems on the PEDro scale was 3.3. This ranged from an average of 2.75 items to four items.

## Post-acquisition test

Ten studies with a total of k = 11 samples were included for the post-acquisition endpoint. In total m = 22 pairwise comparisons with $n = 5$ treatments and d = 5 different designs were available. The distribution of the network is presented in Fig. 3. The network consisted mainly of closed loops, open loops were found for the comparisons control *vs* switching focus" and "holistic focus *vs* switching focus" (*i.e.*, for these comparisons only indirect evidence was available). The most common comparison was "EFA" *vs* "IFA" with m = 10 comparisons. The network plot with the number of comparisons is presented in Fig. 3A.

The network analysis revealed that, in comparison to the control group, "EFA" had the largest effect on performance measures at post-acquisition testing (SMD: 0.99; 95% CI [0.4 to 1.57]; *p*: 0.001). The least effective intervention compared to control was "IFA" (SMD: 0.11; 95% CI [−0.49 to 0.71]; *p*: 0.72). A forest plot of the effectiveness of all interventions at the post-acquisition endpoint with the control intervention set as the comparator is presented in Fig. 4.

The following treatment ranking was obtained based on the P-scores (from highest to lowest): "EFA", "switching focus", "holistic focus", "IFA" and "control" (Appendix A2).

Heterogeneity in this analysis was substantial ($I^2$: 71.3%). Further exploration showed that the heterogeneity could be explained by heterogeneity within designs (*i.e.*, direct

**Table 2 Characteristics of included studies.**

| Study-ID | Design | Sample size and Age | Sport, motor skill trained | Experience of participants | Attentional foci used | Training period and outcome measures | Results, measurements | Results (final values) |
|---|---|---|---|---|---|---|---|---|
| *Abedanzadeh, Becker & Mousavi (2022)* | RCT | N = 60<br>Sex: (60 M)<br>Age: M = 19.56, SD = 0.97 | Badminton, short serve | Novices | EFA: "focus on the movement of the racket during the serve." | Post-acquisition test: 3 × 10 trials over 5 days | Accuracy, score (↑) | PAT<br>EFA mean 3.22 (SD: 0.55)<br>IFA mean 3.01 (SD: 0.76)<br>Holistic mean 3.28 (SD: 0.81)<br>Control mean 2.65 (SD: 0.48) |
| | | | | | IFA: "focus on the movement of your arm during the serve." | 2 days later, retention test with 10 trials | | RT<br>EFA mean 2.38 (SD: 0.46)<br>IFA mean 2.05 (SD: 0.73)<br>Holistic mean 2.47 (SD: 0.35)<br>Control mean 1.87 (SD: 0.3) |
| | | | | | Holistic: "focus on feeling smooth and fluid when completing the serve."<br>Control: no focus cue given | Transfer test: 30min after RT; TT with 10 trials | | TT<br>EFA mean 2.10 (SD: 0.4)<br>IFA mean 1.85 (SD: 0.72)<br>Holistic mean 2.57 (SD: 0.59)<br>Control mean 1.67 (SD: 0.16) |
| *Aiken & Becker (2022)* | RCT | N = 79<br>Sex: (79, 27 M, 52 W)<br>Age: M =19.28, SD = 2.31 | Golf, hipping task | Novices | EFA: "focus on the clubface hitting the bottom of the ball." | Acquisition: 8 × 10 trials | Accuracy, score (ring-score) (↓) | PAT<br>EFA mean 56.33 (SD -)<br>IFA mean 61.23 (SD -)<br>Switching mean 55.54 (SD -) |
| | | | | | IFA: "concentrate on the swinging movement of the arms." | After 24 h, retention test and transfer test 10 trials each | | RT<br>EFA mean 62.34 (SD -)<br>IFA mean 61.55 (SD -)<br>Switching mean 59.02 (SD -) |
| | | | | | Switching: "during preparation: focus on the swinging movement of the arms. During execution: focus on the clubface hitting the bottom of the ball" | | | TT<br>EFA mean 65.03 (SD -)<br>IFA mean 66.14 (SD -)<br>Switching mean 65.51 (SD -) |
| *Al-Abood et al. (2002)* | RCT | N = 16<br>Sex: (16 M)<br>Age: M = 21.3, SD = 1.8 | Basketball, free throw | Novices | EFA (motion effect): "focus on how the model scored a basket." | Pre-test: 5 trials | Performance, score (↑) | PAT<br>EFA mean 0.23 (SD -)<br>EFA-dynamic mean 0.12 (SD -) |
| | | | | | EFA-dynamic (motion dynamics): "Concentrate on the movement patterns of the model." | Post-test: 5 trials | | |

(Continued)

| Study-ID | Design | Sample size and Age | Sport, motor skill trained | Experience of participants | Attentional foci used | Training period and outcome measures | Results, measurements | Results (final values) |
|---|---|---|---|---|---|---|---|---|
| *Benjaminse et al. (2017)* | RCT | N = 90<br>Sex: (45 M, 45 W)<br>Age: M = 23.6, SD = 4.14 | Basketball, evasive manoeuvre | Experts | Visual external: Participants received video feedback on a television screen showing the subject from behind. | 3 sessions, 35 trials per session | Vertical ground reaction force, N/kg (↑) | PAT<br>IFA mean 19.95 (SD: 3.55)<br>IFA mean 19.95 (SD: 3.55)<br>Control mean 21.15 (SD: 3.1) |
| | | | | | Verbal internal: "bend the torso forward, bend the knee and keep the knee straight over the foot." | 1 acquisition test and 2 retention tests (after 1 and 4 weeks). | | RT<br>EFA mean 22.55 (SD: 3.25)<br>IFA mean 20.10 (SD: 3.6)<br>Control mean 21.9 (SD: 3.2) |
| | | | | | Control: only provided with the general instructions | | | |
| *Land, Frank & Schack (2014)* | RCT | N = 20<br>No specification for sex<br>Age: M = 26.8, SD = 5 | Golf, putting task | Novices | EFA: "concentrating on the correct trajectory and speed of the ball." | 3 days acquisition: 180 trials per day | Accuracy, mm (↓) | PAT<br>EFA mean 458.13 (SD: 66.44)<br>IFA mean 711.42 (SD: 207.61) |
| | | | | | IFA: "focusing on the swing of their arms and hands." | After 2 days, RT: 30 trials | | RT<br>EFA mean 462.28 (SD: 78.9)<br>IFA mean 628.37 (SD: 153.64) |
| *Makaruk et al. (2019)* | RCT | N = 60<br>Sex: (120 M)<br>Age: M = 21.7, SD = 1.4 | Football, penalty kick | Novices | EFA: "focus your attention on a specific target." | Acquisition: 12 trials | Accuracy, Cm (↓) | PAT<br>EFA mean 89.82 (SD: 9.13)<br>IFA mean 108.77 (SD: 8.77)<br>Control mean 107.02 (SD: 9.02) |
| | | | | | IFA: "concentrate on the movement of the shooting leg. | | | |
| | | | | | Control: did not receive any attentional-focus instruction | | | |
| *Marchant et al. (2009)* | Cross-over (counterbalanced) | N = 72<br>Sex: (72, 32 M, 40 W)<br>Age: M = 19.82, SD = 3.78 | Dart throwing | Novices | EFA: "aim at the center of the dartboard and throw the dart when it is sharp." | Acquisition: 2 sessions; 50 trials per session, 1 week in between | Accuracy, Score (↓) | PAT<br>EFA mean 3.97 (SD: 0.67)<br>IFA mean 4.59 (SD: 0.7) |
| | | | | | IFA: "concentrate on the movements you have made with each throw." | | | |
| *Munzert, Maurer & Reiser (2014)* | RCT | N = 30<br>Sex: (30, 9 M, 21 W)<br>No age specified (students) | Golf. putting task | Novices | EFA: "concentration on the desired ball path." | Acquisition: 120 trials | Accuracy, cm (↓) | PAT<br>EFA mean 57.8 (SD: 5.19)<br>IFA mean 64.01 (SD: 7.43) |
| | | | | | IFA: "concentration on the execution of a pendulum-like movement." | One day later retention test and transfer test: 20 trials each | | RT<br>EFA mean 60.9 (SD: 10.9)<br>IFA mean 73.5 (SD: 19)<br><br>TT<br>EFA mean 70.3 (SD: 27.8)<br>IFA mean 60.8 (SD: 13.9) |

| Study-ID | Design | Sample size and Age | Sport, motor skill trained | Experience of participants | Attentional foci used | Training period and outcome measures | Results, measurements | Results (final values) |
|---|---|---|---|---|---|---|---|---|
| *Poolton et al. (2006)* (experiment 1) | RCT | N = 30<br>Sex: (30, 7 M, 23 W)<br>Age: M = 24.1, SD = 5,94 | Golf, putting task | Novices | EFA: "focus on the swing of the putter head."<br><br>IFA: "focus attention on the swing of your hands." | Acquisition: 10 blocks, 30 trials per block.<br><br>Directly after the test phase, retention test and transfer test: 30 trials each | Accuracy, total number of successful putts per block of 30 attempts (↑) | PAT<br>EFA mean 8.6 (SD -)<br>IFA mean 8.15 (SD -)<br><br>RT<br>EFA mean 9.79 (SD -)<br>IFA mean 9.63 (SD -)<br><br>TT<br>EFA mean 9.79 (SD -)<br>IFA mean 7.29 (SD -) |
| *Poolton et al. (2006)* (experiment 2) | RCT | N = 39<br>Sex: (39, 15 M, 24 W)<br>Age: M = 20.4, SD = 3.84 | Golf, putting task | Novices | As in experiment 1 but with secondary task load *e.g.*, EFA: "hit the ball with the center of the of the club head."<br><br>As in experiment 1 but with secondary task load *e.g.*, IFA: "hold wrists firmly." | Acquisition: 10 blocks, 30 trials per block.<br><br>Directly after the test phase, retention test and transfer test: 30 trials each | Average score per block of 30 attempts (↑) | PAT<br>EFA mean 43.96 (SD -)<br>IFA mean 35.18 (SD -)<br><br>RT<br>EFA mean 43.59 (SD -)<br>IFA mean 44.27 (SD -)<br><br>TT<br>EFA mean 26.96 (SD -)<br>IFA mean 31.78 (SD -) |
| *Rostami et al. (2020)* | RCT | N = 32<br>Sex: (32 W)<br>Age: 18–24 | Volleyball, landing after a block | Novices | EFA: Verbal and visual instruction with the goal of improving movement technique, reduce landing forces, and improve movement patterns.<br><br>Control group: | 18 training sessions: 3 times per week for 6 weeks | Distance, cm (↑) | PAT<br>EFA mean 458.66 (SD: 45.24)<br>C mean 398.41 (SD: 40.81) |
| *Wulf, Lauterbach & Toole (1999)* | RCT | N = 22<br>Sex: (22, 13 M, 9 W)<br>Alter: 21–29 | Golf, pitching task | Novices | EFA: "concentration on the club movement."<br><br>IFA: "concentration on your body movements (arms)." | Acquisition: 80 trials<br><br>1 day later, retention test 30 trials | Accuracy, score (↑) | PAT<br>EFA mean 23.57 (SD -)<br>IFA mean 13.64 (SD -)<br><br>RT<br>EFA mean 22.62 (SD -)<br>IFA mean 13.38 (SD -) |
| *Yamada, Raisbeck & Porter (2021)* | RCT | N = 42<br>No specification for sex<br>Age: M = 21.74, SD = 2.26 | Standing long jump | Novices | EFA (cone present): "focus on jumping as close as possible to the cone."<br><br>Imagined EFA (cone imagined): "focus on jumping as close as possible to the imaginary cone | Acquisition: 4 blocks, 2 trials per block.<br><br>After 24h, RT and TT: 2 trials | Distance, cm (↑) | PAT<br>EFA mean 211.3 (SD: 12.78)<br>IEFA mean 214.78 (SD: 13.0)<br><br>RT<br>EFA mean 207.17 (SD: 11.74)<br>IEFA mean 209.78 (SD: 11.52)<br><br>TT<br>EFA mean 206.96 (SD: 11.74)<br>IEFA mean 211.74 (SD: 11.52) |

**Note:**
M (men), W (women). Control group (C). External focus of attention (EFA). Imagined external focus of attention (IEFA) Internal focus of attention (IFA). Post-acquisition test (PAT). Retention test (RT). Transfer test (TT). Higher values indicate better performance: ↑; Lower values indicate better performance: ↓.

comparisons with Q: 32.59 and *p*: 0.001). The design with the highest contribution to the heterogeneity was "EFA" *vs* "IFA" (Q: 18.27 and *p*: 0.003). Between designs the inconsistency was not statistically significant (Q: 8.31 and *p*: 0.14). A netleague table comprising the direct effect estimates and the network estimates is presented in Appendix A3. The largest differences were observed for the comparisons with only one study presenting direct evidence (*i.e.*, "IFA" *vs* "holistic focus" and "IFA" *vs* "switching focus"). In addition, a heat plot showed that overall, the inconsistency was small in the analysed network (Appendix A4). Most studies were symmetrical distributed in a comparison adjusted funnel plot (Appendix A5).

Two studies reporting on different versions of an EFA at post-acquisition and could not be integrated into the network meta-analysis. *Yamada, Raisbeck & Porter (2021)* demonstrated that an imagined EFA and an EFA yielded comparable results for this endpoint. *Al-Abood et al. (2002)* showed similar effectiveness of a dynamic EFA compared to an EFA.

### Retention test

Seven studies (k = 7) were included for the retention test endpoint. In total, m = 13 comparisons with *n* = 5 treatments and d = 4 designs were available in the dataset. The network is presented in Fig. 3B. The network comprised mainly of closed loops. For the comparisons "switching focus *vs* control" and "holistic focus *vs* switching focus" only indirect evidence was available. With regard to the remaining comparisons, at least one study provided direct evidence. The most frequent comparison was "EFA" *vs* "IFA" with m = 7 comparisons.

The results of the network meta-analysis showed that the intervention with the largest effect on performance measures at retention testing was "holistic focus" with a moderate to large effect size (SMD: 0.75; 95% CI [−0.1 to 1.6]; *p*: 0.08) followed by "EFA" with a moderate effect size (SMD: 0.56; 95% CI [0.20–0.92]; *p*: 0.002) when compared to IFA (Fig. 5). P-Scores indicated the following treatment ranking at retention testing: "holistic focus" (P-Score: 0.75), "EFA" (P-Score: 0.63), "switching focus" (P-Score: 0.57), "control" (P-Score: 0.5), "IFA" (P-Score: 0.06). Heterogeneity was moderate in this network meta-analysis ($I^2$: 44.5%). Heterogeneity within designs was not statistically significant (Q: 3.72 and *p*: 0.29). Similar results were found for between designs inconsistency (Q: 7.22 and *p*: 0.06). A comparison of direct evidence and network estimates is presented in Appendix A6. Overall network and direct evidence estimates were similar. Net splitting did not show any significant differences between direct and indirect effect estimates (Appendix A7). A comparison adjusted funnel plot did not show asymmetry (Appendix A8).

### Transfer test

The network meta-analysis for the transfer endpoint comprised k = 4 studies, m = 8 comparisons, *n* = 4 interventions, and d = 3 designs. The network plot is presented in Fig. 3C. The network was comprised of closed loops, with the exception of the comparison between a holistic focus and a switching focus. Most comparisons were found for the comparison "EFA" *vs* "IFA (m = 4).

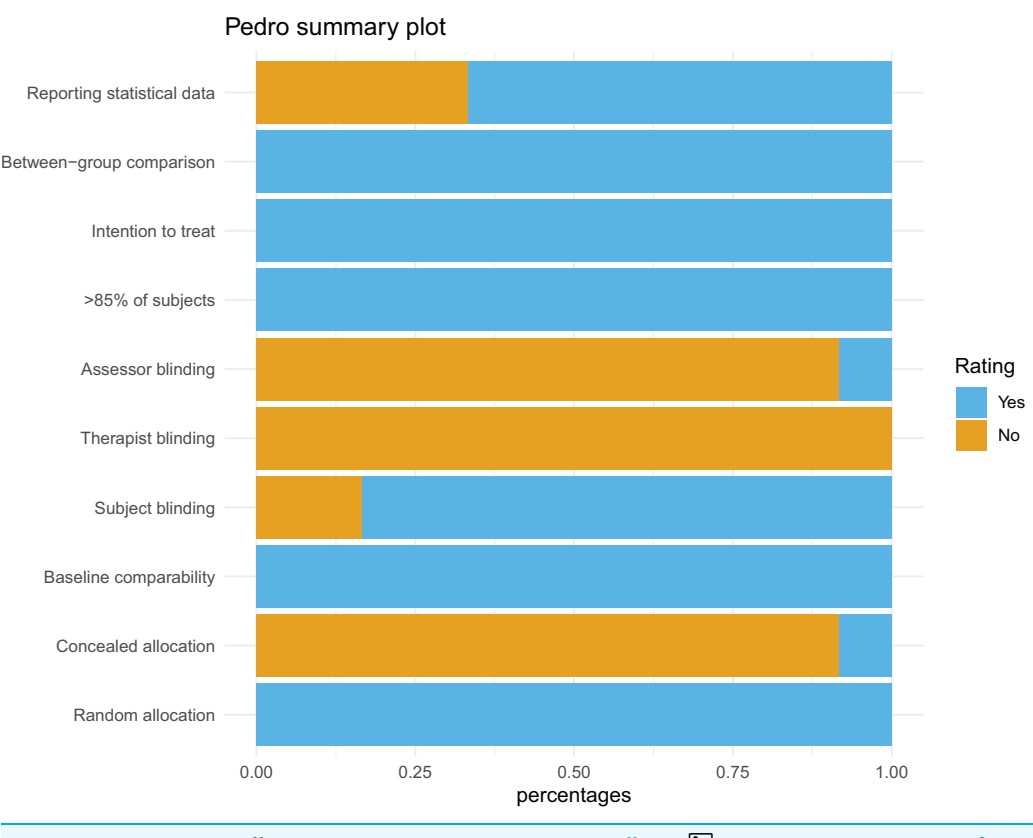

**Figure 2  PEDro overall assessment.**

The most effective intervention at transfer testing (*i.e.*, in comparison to IFA) was "holistic focus" with a large effect size (SMD: 1.16; 95% CI [0.47–1.86]; *p*: 0.001), followed by an "EFA" (SMD: 0.35; 95% CI [0.02–0.68]; *p*: 0.04) (Fig. 6). The following treatment ranking was obtained: "holistic focus" (P-Score: 0.99), "EFA" (P-Score: 0.57), "switching focus" (P-Score: 0.35) and "IFA" (P-Score: 0.08).

Heterogeneity in the network meta-analysis at transfer testing was classified as low ($I^2$: 0%). Heterogeneity within and between designs was not statistically significant (Q: 0.37 and *p*: 0.54 respectively Q: 1.53 and *p*: 0.46). A netleague table presenting direct and network estimates is presented in Appendix A9. Net splitting did not show any statistically significant differences between direct and indirect effect estimates (Appendix A10).

## Sensitivity analyses

In order to investigate the heterogeneity at the post-acquisition endpoint several analyses were performed. The majority of heterogeneity was identified in the comparison EFA *vs* IFA. We explored the influence of the following potential moderator variables: age of the participants, number of trials during acquisition and methodological quality with the number of points on the PEDro scale. In examining the influence of age on the post-acquisition endpoint, the meta-regression analysis yielded an $R^2$ value of 0%, indicating that age did not account for any of the variance in the effect sizes across studies. The model's *p*-value was 0.48, suggesting that age was not a statistically significant

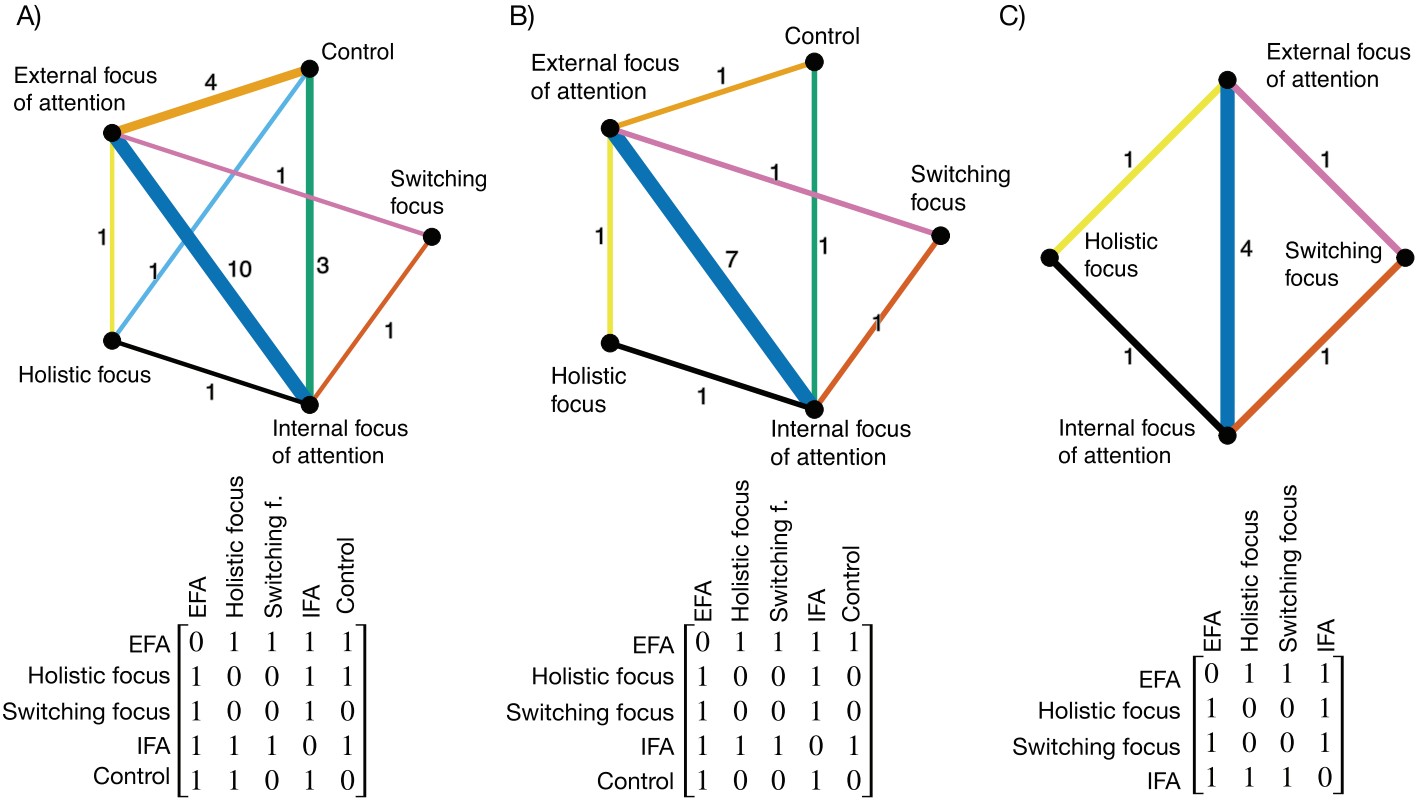

**Figure 3** (A) Network plot—post-acquisition test; (B) Network plot—retention test, (C) Network plot—transfer test. NB. The numbers indicate the number of pair-wise comparisons. The following colours were used to indicate comparisons: orange: EFA *vs* control, yellow: EFA *vs* holistic focus, blue: EFA *vs* IFA, black: holistic focus *vs* IFA, brown: IFA *vs* switching focus, lilac: EFA *vs* switching focus. The network is tabulated below each plot in a matrix, with 0 indicating no direct connection between two interventions and 1 indicating a direct connection. Switching f.: switching focus.                                                           

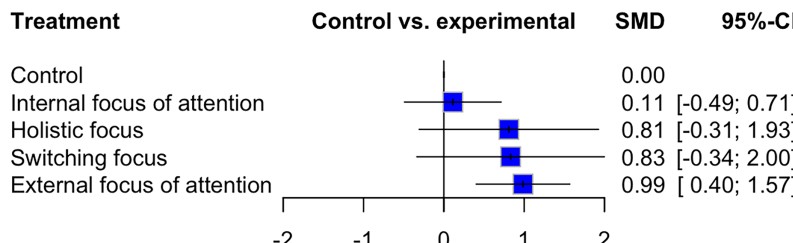

**Figure 4** Forest plot—post-acquisition test. NB. The control intervention was set as the comparator.

predictor of the outcomes. The estimated effect size for age was 0.07, with a 95% confidence interval ranging from [−0.176 to 0.33]. Similar results were found for the variables number of trials (model: $R^2$: 0%, $p$: 0.48, effect estimate: −0.001 (95% CI [−0.005 to 0.003]) and methodological quality (model: $R^2$: 0%, $p$: 0.53, effect estimate: 0.33 (95% CI [−0.86 to 1.52]).

Another sensitivity analysis was performed in an attempt to decrease heterogeneity within the post-acquisition endpoint network meta-analysis. Within this analysis we

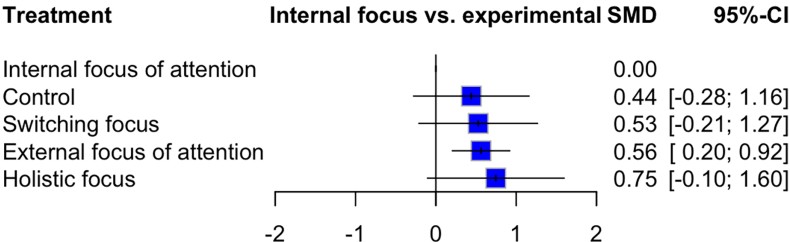

**Figure 5 Forest plot—retention test.** NB. IFA was set as the comparator.

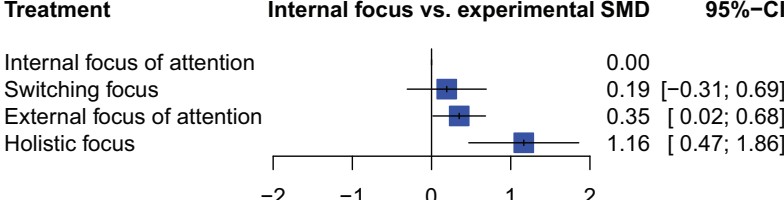

**Figure 6 Forest plot—transfer test.** NB. IFA was set as the comparator.

excluded a single study from the network analysis. The study of *Wulf, Lauterbach & Toole (1999)* showed a very large effect (SMD: 2.51; SE: 0.58) in favour of an EFA and the effect estimates differed considerably from the other studies in the analysis. The exclusion of this study resulted in a slight reduction in heterogeneity ($I^2$: 66% *vs* 73%) (Appendix A11).

# DISCUSSION

Moving from the quantitative results to a more detailed discussion, we first address the overall findings and then discuss each study endpoint (*i.e.*, post-acquisition, retention and transfer). The aim of the present systematic review was to investigate the effectiveness of different attentional foci on performance and learning of a sport-specific motor task in individuals practising a sport. For all three endpoints, we analysed a greater effectiveness of an EFA and holistic focus compared to an IFA.

For performance of motor skills at post-acquisition, an EFA intervention demonstrated a large effect size in enhancing motor skills compared to a control intervention or an IFA. The precise magnitude of the effect remains uncertain, as the confidence interval suggests it could range from small to moderate, up to a possibility of being very large. Although modified EFA interventions (imagined or dynamic EFA) were identified, the effects of these interventions were not subjected to quantitative investigation.

Regarding performance at a retention test, the IFA intervention was analysed as having the lowest effectiveness. The intervention with the highest effectiveness at this endpoint was a holistic focus. However, confidence in this estimate is low due to a large confidence interval, which is also compatible with no superiority compared to IFA. The EFA instruction yielded relatively robust results, indicating a moderate effect with an estimated confidence interval that spans from small to large.

The intervention with the highest effect size for the transfer test endpoint was a holistic focus. The confidence interval was compatible with a moderate to very large effect. An EFA was also more effective than an IFA at the transfer test. However, the effect size was small, with the potential for the observed effect to range from zero to a large magnitude.

Following the presentation of the findings of each study endpoint, several promising attentional focus interventions are discussed in the following section. The largest effects were analysed for a holistic focus. However, only one study (*Abedanzadeh, Becker & Mousavi, 2022*) used this intervention and therefore there remains uncertainty about the effectiveness.

Modified versions of the EFA were the imagined and the dynamic EFA. Both were only explored in single studies (*Al-Abood et al., 2002*; *Yamada, Raisbeck & Porter, 2021*) and should therefore be investigated in further follow-up studies that directly compare them.

A switching focus (*i.e.*, an IFA during movement preparation and an EFA during execution) was analysed as having a high effect size at post-acquisition, moderate at retention and small at transfer testing. All three estimates were associated with large uncertainty, which was caused by only one study reporting on this intervention. However, within all analyses the effect size of a switching focus was higher than an IFA instruction. This may indicate that the detrimental effects of an IFA may be (partly) avoided if during movement execution the attentional focus is directed externally. Further exploration of this mechanism may be of importance for practitioners. In certain cases, teaching a motor skill effectively using only an EFA may be challenging. For example, rehabilitation specialists were observed to utilise an IFA more frequently than an EFA (*Johnson, Burridge & Demain, 2013*), even though evidence suggests that an EFA is more effective.

A potential explanation for the higher effectiveness of EFA compared to IFA is the constrained action hypothesis, which suggests that an IFA interferes with motor skill acquisition by constraining natural processes (*Wulf, McNevin & Shea, 2001*). As very well-known in the field of cognitive sciences, the body has a double significance. Indeed, from a side, it is a physical object mediating all the interactions we have with the external environment through the senses and movements. From the other side, the body is itself an object of perception, whose current status is transmitted to the brain *via* multiple bodily signals (*Risso & Bassolino, 2022*). It is likely that in the former case (EFA), more cognitive resources are available because the body is processed as a means of interaction through automatic processing characterized by faster and more reflexive adjustments, at least in healthy participants. Conversely, when the body is at the center of the attentional focus, and therefore itself an object of perception, this may represent an additional cognitive burden affecting automatic processing.

Within this section inconsistency and heterogeneity within the network analyses is discussed. In order to explore the origin of the inconsistency at post-acquisition a series of sensitivity analyses were conducted. The principal source of inconsistency was identified in the comparison between the EFA and IFA. Meta-regressions were utilised to identify the influence of the variables "age", "number of trials" and "methodological quality". None of these variables had a significant influence on the effect estimates. A second sensitivity analysis was performed in order to reduce inconsistency in the post-acquisition network.

The omission of one study (*Wulf, Lauterbach & Toole, 1999*) with a particular large effect size in favour of an EFA reduced heterogeneity by seven percentage points. However, we could not identify any practical variables which were considerable different between this study and the others. Specifically, the variables skill level "novices", trained motor skill "golf putting task" and sample size were comparable to the other studies. Therefore, we recognise that other not explored factors are likely to influence the effectiveness of the different attentional foci. These might be of methodological or practical origin and future studies are necessary to identify these.

Following the discussion of inconsistency, we provide a brief contextualisation of our findings in relation to other systematic reviews in this field. Two previous systematic reviews were identified, which investigated the effectiveness of different attentional foci on motor skill acquisition. Both systematic reviews reported a pairwise comparison between an EFA and an IFA. *Chua et al. (2021)* analysed the effectiveness in a very broad population. The authors reported that an EFA was in general more effective compared to an IFA. Similar results were found in our analysis. However, at post-acquisition testing, our effect estimate was considerably larger in favour of an EFA. This larger effect might be explained by the more homogenous population in our study, which focus on healthy people training a sport-specific motor skill.

A second systematic review investigating the effectiveness of an EFA *vs* an IFA was published by *Makaruk, Starzak & Porter (2020)*. In this review, the population was more specific and focused on healthy adults and jumping performance was analysed. The differences between our results (*i.e.*, a large effect in this review) might be explained by differences in selection criteria. Some of the studies included in the review by *Makaruk et al. (2019)*, were not included in our systematic review. We excluded studies i) if the allocation method was not stated as "randomized" and ii) if participants received both interventions (*e.g.*, the first intervention was EFA and the second IFA), and the period between the two interventions had to be specified and sufficiently long to avoid carry-over effects.

## Limitations

The results of the present study should only be interpreted with caution due to the heterogeneity in the analyses. This was especially the case for the analysis of the post-acquisition endpoint. There are several possible reasons for the moderate to considerable heterogeneity: i) the variability of the sample sizes of the included studies; ii) the differences in the population in the included studies, such as different sports or different skill levels, various task complexity and motor skills practiced; iii) methodological heterogeneity. However, we tried to eliminate this source of heterogeneity with strict selection criteria. Only studies with randomised controlled trial design were included in this systematic review. In addition, the analyses of the methodological quality showed similar ratings on the PEDro items and most studies had lower ratings in the same categories.

Another limitation of this review was the limited sample size of the included studies. Most of the studies had relatively small sample sizes and a small study bias cannot be

excluded (*Sterne, Egger & Smith, 2001*). Furthermore, the data available did not permit an assessment of the task-specific effects.

Future research should consist of larger study samples, investigating new attentional focus strategies, and clarifying the optimal timing and combination of different foci during skill acquisition and motor learning. In addition, further research should focus on the strategies identified, such as holistic focus and imagined EFA, using larger sample sizes and randomised controlled trial designs. In practical terms, the current evidence strongly supports the recommendation of using an EFA for individuals training in specific sports.

### Implications

The present study has several implications for research. First, our results indicate that despite a large body of evidence, there is still a lack of larger studies (*i.e.*, recruiting more than 100 participants) investigating this motor learning principle. Second, currently, studies using a cross-over design often do not specify the time interval between two different training periods (*i.e.*, the "wash-out period"), which limits any interpretation due to possible carry-over effects. Moreover, our findings supporting the promising effects obtained with specific attentional strategies such as "holistic focus" or "imagined EFA" motivate further follow-up studies of these methods in sports, but also in other populations (*e.g.*, neurological patients).

The implications for practice are as follows: crucially, the present findings indicate that there is sufficient evidence to recommend the use of an EFA for the acquisition of sport-specific motor tasks. An IFA showed lower effectiveness compared to an EFA on all three endpoints and should therefore only be used with caution in this specific population (*i.e.*, healthy adults training a specific sport).

## CONCLUSIONS

This systematic review with network meta-analysis investigated the effectiveness of different attentional foci on the performance and learning of sport-specific motor tasks in healthy adults. The findings showed consistent evidence supporting the superiority of an EFA over an IFA across different motor skill acquisition scenarios. The study's approach, including a network meta-analysis and a diverse range of sports and attentional focus strategies, enhances the reliability and applicability of the findings. The analysis identified the potential benefits of new attentional focus strategies, such as the holistic focus and switching focus, supplying valuable data for researchers and practicians in the field of motor skill acquisition and motor learning. However, uncertainty for the effectiveness of these interventions is large and they should be used with caution. Another caveat is that a considerable inconsistency was identified in analysis of the post-acquisition endpoint. Despite these limitations, the current evidence supports the recommendation of using an EFA for individuals training in specific sports.

### Funding

The authors received no funding for this work.

### Competing Interests

The authors declare that they have no competing interests.

### Author Contributions

- Emmanuel Favre-Bulle conceived and designed the experiments, performed the experiments, analyzed the data, prepared figures and/or tables, authored or reviewed drafts of the article, and approved the final draft.
- Siri Nyfeler conceived and designed the experiments, performed the experiments, analyzed the data, prepared figures and/or tables, authored or reviewed drafts of the article, and approved the final draft.
- Chloé Schorderet conceived and designed the experiments, performed the experiments, analyzed the data, prepared figures and/or tables, authored or reviewed drafts of the article, and approved the final draft.
- Gaia Risso conceived and designed the experiments, analyzed the data, authored or reviewed drafts of the article, and approved the final draft.
- Michela Bassolino conceived and designed the experiments, analyzed the data, authored or reviewed drafts of the article, and approved the final draft.
- Karl Martin Sattelmayer conceived and designed the experiments, performed the experiments, analyzed the data, prepared figures and/or tables, authored or reviewed drafts of the article, and approved the final draft.

### Data Availability

This is a systematic review/meta-analysis.

### Supplemental Information

Supplemental information for this article can be found online at http://dx.doi.org/10.7717/peerj.17799#supplemental-information.

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
