# Peer review of "The effectiveness of different attentional foci on the acquisition of sport-specific motor skills in healthy adults: a systematic review with network meta-analysis"

_PeerJ, doi:10.7717/peerj.17799_

## Round 0.1 · original submission · Major Revisions

Dear authors:

After receiving the reviews of his manuscript entitled: "The effectiveness of different attentional focus on the acquisition of sport-specific motor skills in healthy adults: a systematic review with network meta-analysis," we have considered that some changes are required. Please pay attention to the reviewers' comments.

Cordially,
Dr. Manuel Jimenez

·

Basic reporting

The authors adequately provide a thorough literature review from a historical context. They do not provide a hypothesis.

Experimental design

A network meta-analysis is uncommon and more explanation is necessary to explain to the reader the purpose and benefit of the network variety and how its interpretation is different from that of a typical meta-analysis.

I do question the rationale for separating the dynamic and imagined external foci of attention from regular external focus of attention. These are endless varieties to an EF, not all of which deserve their own categories.

Validity of the findings

Given that there was only 1 study using an imagined EF and 1 study using a switching focus and 1 with a holistic focus, it's unable to draw any robust conclusions based on these manipulations. The authors don't necessarily overstate their conclusions, but it raise more questions as to why the authors chose to separate out these external foci of attention manipulations given how thin the evidence is.

Additional comments

April 9, 2024
Dear authors,
I recently had opportunity to review your manuscript “The effectiveness of different attentional foci on the acquisition of sport-specific motor skills in healthy adults: a systematic review with network meta-analysis.” Your manuscript is fairly well written and relatively easy to follow. My biggest concern is that it doesn’t add anything to what is already known. As stated in the manuscript, there have been 2 systematic reviews already and the literature hasn’t advanced that much since 2021. I have supplied further specific comments below that I hope are helpful.

Specific comments:
-Abstract results—More information is needed. Specifically, what are the comparators for these SMDs? Please provide p values.
-Lines 52-54—Avoid quoting directly. Suggest paraphrasing in your own words to better fit the flow of the paragraph.
-Line 183—Why not include all outcomes for each study?
-Lines 210-219—A network M-A isn’t as common as a regular M-A. Suggest spending time to explain what a network M-A is, its assumptions, why it is useful, and how to interpret.
-Line 218—Better define what ‘sufficiently similar’ is.
-Line 275—Please define ‘transitivity.’
-Line 286—Please briefly define open-loop and closed-loop comparisons.
-Line 296 and throughout—Please clarify what the comparator is for each of these SMDs.
-Lines 358-366—Claims of statistical significance are not warranted. You have provided no p values for any of these comparisons. A statistically significant effect is not the same thing as a moderate effect size. You can establish an a priori significance value (i.e., statistically significant, or p value) or an a priori effect size. I would change the language throughout the discussion to reflect effect size and not statistical significance.
-Lines 376-388—You have already discussed these papers in the introduction. I would shorten the discussion here.
-Limitations—What about different tasks? The benefit of the network M-A is that you can assess different tasks, but no press is given to this. If there is not enough data to assess task-specific effects, this should be in the limitations.
-‘Imaged’ should be ‘imagined’ throughout.
-It’s not clear why the authors are splitting imagined and dynamic from regular EFA. This seems arbitrary. There are also near and far EFAs—why were these not split off? There are endless variations on external focus and imagined / dynamic should simply be considered EFA.
-Lines 432-436—This is discussion content, not appropriate for conclusion.
-Lines 440-441—Not conclusion content. Should be in discussion.
-Line 448—There was no discussion of the switching focus. If it is in the conclusion paragraph, there should be discussion of what this means and how to apply.
-Lines 450-456—This belongs in the limitations section.
-Forest plots—There should also be an aggregate point estimate at the bottom of each plot.

Reviewer 2 ·

Basic reporting

The manuscript offers a robust network meta-analysis that significantly contributes to our understanding of how attentional focus affects motor skill acquisition. By incorporating various sports and a range of attentional strategies, the value of the manuscript is markedly enhanced. However, it's vital to address the detailed statistical issues, clarify terminology, and improve visual presentations to boost the manuscript’s impact and dependability.

Experimental design

NA

Validity of the findings

The conclusions drawn about the superiority of EFA over IFA are based on significant results in some comparisons but not all. Given the noted substantial heterogeneity and the wide confidence intervals in some results, it would be prudent to state these conclusions with caution, recognizing the variability and potential influence of unexplored factors.

Additional comments

1. There's an inconsistency in the confidence intervals reported for the 'imagined EFA' at the post-acquisition endpoint. The interval ranges from -1.0 to 2.59, crossing zero and thus, implying no significant effect. Please reevaluate these confidence interval calculations to ensure their accuracy, as this has significant implications for interpreting effectiveness.
2. The manuscript notes substantial heterogeneity (I2: 71.3%) in the network meta-analysis for the post-acquisition endpoint. It’s crucial to delve into potential sources of this heterogeneity in more detail, considering subgroup analyses or meta-regression to assess the impacts of moderators such as study quality, participant characteristics, or the specifics of interventions.
3. The network plots (Figure 3a, 3b, 3c) require enhancements for better readability. Consider employing varied colors or shapes for different interventions and make sure that the labels are legible and not overlapping, aiding readers in more effectively interpreting the results.
4. The terms 'EFA' and 'external focus' are used interchangeably; it would be beneficial to standardize the terminology throughout the manuscript.
5. There is a typographical error in the phrase 'Data was data was not available in numeric form' (line 199) that needs correction.
6. The usage of 'imagined' versus 'imaged' EFA varies; standardizing these terms would aid in maintaining consistency throughout the document.
7. The logical flow between the results and discussion sections could see improvement. The discussion section jumps into detailed analysis without a clear linkage back to the specific results. Enhancing the transition sentences to clearly outline which findings from the results section are being discussed would help maintain a logical flow and coherence.

---

## Round 0.2 · accepted · Accept

Dear Co-Authors:

We are pleased to inform you that your manuscript titled “The Effectiveness of Different Attentional Foci on the Acquisition of Sport-Specific Motor Skills in Healthy Adults: A Systematic Review with Network Meta-Analysis” has been accepted for publication in PeerJ.

After a thorough review by our editorial committee and expert reviewers, we have concluded that your work meets the high standards of quality and scientific relevance required by our journal. We particularly appreciate the depth of your analysis and the significant contribution it makes to the field of motor skill acquisition in sports and the use of different attentional foci.

Regards

Dr. Manuel Jiménez

·

Basic reporting

See previous review.

Experimental design

See previous review.

Validity of the findings

See previous review.

Additional comments

The authors have diligently responded to all of my previous comments. The addition of information explaining the purpose and use of a network meta-analysis goes a long way to boost the necessity of the present manuscript.

Reviewer 2 ·

Basic reporting

NONE

Experimental design

NONE

Validity of the findings

NONE